

# B-flood 1.0: an open-source Saint-Venant model for flash flood simulation using adaptive refinement

Geoffroy Kirstetter[1], Olivier Delestre[1], Pierre-Yves Lagrée[2], Stéphane Popinet[2], and Christophe Josserand[3]

[1]Laboratoire de Mathématiques J.A. Dieudonné - Polytech Nice-Sophia , Université de Nice - Sophia Antipolis, CNRS, UMR 7351, Parc Valrose, 06108 Nice cedex 02, France
[2]Sorbonne Université, CNRS, UMR 7190, Institut Jean Le Rond d'Alembert, F-75005 Paris, France
[3]Laboratoire d'Hydrodynamique (LadHyX), UMR7646 CNRS-Ecole Polytechnique, 91128 Palaiseau CEDEX, France

**Correspondence:** Kirstetter Geoffroy (geoffroy.kirstetter@gmail.com)

**Abstract.** The French Riviera is very often threatened by flash floods. These hydro-meteorological events, which are fast and violent, have catastrophic consequences on life and properties. The development of forecasting tools may help to limit the impacts of these extreme events. Our purpose here is to demonstrate the possibility of using b-flood (a subset of the Basilisk library http://basilisk.fr/) which is a 2D tool based on the shallow water equations and adaptive mesh refinement. The code

is first validated on analytical test cases describing different flow regimes. It is then applied on the Toce river valley physical model produced by ENEL-HYDRO in the framework of the CADAM project and on a flash flood case over the urbanized Toce produced during the IMPACT project. Finally, b-flood is applied on the flash flood of October 2015 on Cannes city in south-east France, which demonstrates the feasibility of using a software based on the shallow water equations and mesh refinement for flash flood simulation on small watersheds (less than $100\text{km}^2$) and on predictive computational time scale.

# 1   Introduction

The south of France is very often affected by flash floods, strong and rapid events that arise particularly in the summer and autumn due to slow moving convective storms bringing moisture from the Mediterranean sea, with the induced rainfall amplified by topographic influences (Sene, 2012). Some big and catastrophic flash floods occurred in June 2002 in the Gard region (Delrieu et al., 2005), June 2010 around Draguignan city (Javelle et al., 2014) and more recently in October 2015 in the French

Riviera (Carrega, 2016; Saint-Martin et al., 2018), particularly affecting the city of Cannes. Watersheds located in the French Riviera are steep and cover generally less than $100\text{km}^2$, which induces short hydrological responses (a few minutes to a few hours). On these watersheds, two types of flood can be defined (Organization, 2011): riverine flood which is encountered in the upstream part of the river basin and urban flood which occurs in the downstream part of the watershed. Most of these watersheds are densely urbanised and this density is increasing in time (Fox et al., 2019), so that people's lives, properties

(Carrega, 2016; Saint-Martin et al., 2018) and even health (Jacq et al., 2016) are highly threatened by these hazardous climatic events. The simulation of such disastrous climatic events therefore appears to be a key and crucial objective from public safety to urban planning but also because of its important economic consequences. In particular, shorter than real-time simulations



allowing for practical predictions on the storm location and the subsequent flood propagation represent a major challenge for the numerics. In this context of real-time forecasting, there exists three main categories of flood mapping methodologies for the upstream part. The first methodology consists in using the 2D shallow water equations or simplifications (kinematic wave, diffusive wave and local inertia approximations) and/or parallelized (CPU or GPU) resolution algorithm to speed up

calculation times (Lisflood, Iber, Telemac, ...). This has been first applied for mapping large rivers at a continental scale and at fairly high resolutions (Pappenberger et al., 2012; Alfieri et al., 2014; Sampson et al., 2015; Dottori et al., 2016). These methods are gradually evolving towards more local applications and finer resolutions (meters) in order to map floods on small rivers (Cea and Bladé, 2015; Xia et al., 2017; García-Feal et al., 2018; Nguyen et al., 2016; Neal et al., 2018; Sanders and Schubert, 2019). The second flood mapping methodology is based on the application of 1D hydrodynamic models. It is based

on the extraction of cross-sections from a Digital Terrain Model (DTM) (Choi and Mantilla, 2015; Le Bihan et al., 2017; Lamichhane and Sharma, 2018). The third methodology consists in a direct infilling of the DTM from a locally determined water level. This group includes the AutoRoute method (Follum et al., 2017), as well as several approaches based on the concept of height above the nearest drainage point (HAND or Height Above Nearest Drainage) (Rennó et al., 2008; Nobre et al., 2011): f2HAND (Speckhann et al., 2018), Geoflood (Zheng et al., 2018), MHYST (Rebolho et al., 2018). With this approach, a

flow/height relationship is determined from the geometry of the cross-section extracted from the DTM (averaged over a section for HAND-based approaches) using a hydraulic formula (Manning Strickler, Debord, ...). These approaches have the advantage of being very efficient in computing time (Teng et al., 2017) although their accuracy limits have already been highlighted compared to conventional 2D approaches (Afshari et al., 2018). Recently, (Hocini et al., 2020b, a) propose a comparison of mapping methods from each of these families in the context of flash floods observed in south-eastern France. This work has

shown that the 2D method was more accurate. Moreover, most of the operational flood forecasting systems are based on a coupling of a rainfall-runoff/hydrological model and a routing/hydraulic model as reviewed in Jain et al. (2018). Concerning the downstream part of the domain which is densely urbanised (Saint-Martin et al., 2018; Fox et al., 2019), because of the complex geometry of the city it is recommended to use 2D modeling (Mignot et al., 2019). For these reasons, we have chosen to use a fully-integrated model based on the 2D shallow water equations with rain source terms solved thanks to a finite volume

scheme on an adaptive meshes using the software Basilisk (http://basilisk.fr/). The originality of the model is to gather in the same code both the fluvial and urban flood configurations using up-to-date numerical schemes with (Buttinger-Kreuzhuber et al., 2019; Horváth et al., 2020) a forecasting computation time, thanks to the automatic adaptive mesh refinement.

In the first part we present the model, the numerical method and the adaptive mesh technique. In the second part we first exhibit the ability of the Basilisk software to catch different types of flow regimes for analytical solutions developed in (Mac-

Donald et al., 1997) and implemented in the SWASHES library (Delestre et al., 2013). We then apply Basilisk for the Toce river valley physical model (Valiani et al., 1999) produced by ENEL-HYDRO (ex ENEL CRIS) laboratories in Milan in the framework of the CADAM project (Concerted Action on Dam break Modeling, (Morris, 2000)). This is a physical model at scale 1:100 of a submersion wave on part of the Toce river valley in the occidental Alps, Italy. The irregularities in the domain give birth to coexistence of sub-critical flow with super-critical flow. This allows to verify that Basilisk is able to catch properly

the dynamics of the riverine flood encountered in the upper part of the watershed. Basilisk is then used on the urbanized Toce





produced in the framework of project IMPACT (Testa et al., 2007). It shows its ability to reproduce correctly the dynamics of urban flood met in the downstream part of the watershed. This model represents a urban district where buildings are modeled by concrete blocks put at the upstream part of the Toce river case. Finally, b-flood is applied on the flash flood of the 3 October 2015 on Cannes city in south-east France.

## 2  Numerical Scheme

Floods have horizontal length scales much larger than the vertical one. This observation is used as an hypothesis for the model. This gives a pressure which is hydrostatic as a first approximation. Integrating the Navier-Stokes equations over the thin flow depth then gives the following classical Saint-Venant equations (de Saint-Venant, 1871):

$$\partial_t h + \partial_x q_x + \partial_y q_y = S_h, \tag{1}$$

$$\partial_t q_x + \partial_x \left( \frac{q_x^2}{h} + \frac{g}{2} h^2 \right) + \partial_y \frac{q_x q_y}{h} = -gh \partial_x z_b + S_x, \tag{2}$$

$$\partial_t q_y + \partial_x \frac{q_x q_y}{h} + \partial_y \left( \frac{q_y^2}{h} + \frac{g}{2} h^2 \right) = -gh \partial_y z_b + S_y, \tag{3}$$

where $h$ is the local flow depth and $q_x$, $q_y$ are the two components of the horizontal depth-averaged flow rate, $z_b$ the topography, $S_h$ is the mass source term responsible for rainfall and infiltration, and $S_x$ and $S_y$ are the two components of the friction terms on the topography. See De Vita et al. (2020) for a discussion of the loss of details in the transverse integration and Popinet (2020) for non hydrostatic corrections.

The shallow-water equations ((1), (2) and (3)) can be written in conservative, vector form as

$$\frac{\partial}{\partial t} U(h,q) + \frac{\partial}{\partial x} F_x(h,q) + \frac{\partial}{\partial y} F_y(h,q) = S_{zb}(h,q,\cdot) + S(h,q,\cdot) \tag{4}$$

where $U = \begin{pmatrix} h \\ q_x \\ q_y \end{pmatrix}$ is the vector of the conserved variables, $F_x = \begin{pmatrix} q_x \\ \frac{q_x^2}{h} + \frac{g}{2} h^2 \\ \frac{q_x q_y}{h} \end{pmatrix}$ and $F_y = \begin{pmatrix} q_y \\ \frac{q_x q_y}{h} \\ \frac{q_y^2}{h} + \frac{g}{2} h^2 \end{pmatrix}$ the flux, $S_{zb}$ the gravity source term and $S$ for the other source terms. This system of equations is solved thanks to a finite volume approach on a square grid.

### 2.1  Time-Step and time advance algorithm

We use a predictor-corrector scheme as the time-stepping algorithm: it is second-order in time while the source terms are dealt with a time-split method, see "Additional source term" section 2.3 below. We can then so far re-write the equation 4 without the source terms as:

$$\frac{\partial}{\partial t} U = F(U) \tag{5}$$


where $f(U)$ is a numerical flux. Thus, the two steps of the predictor-corrector algorithm can be written for the time step $n+1$ as :

$$U^{n+1/2} = U^n + \frac{\Delta t}{2} F(U^n), \tag{6}$$
$$U^{n+1} = U^n + \Delta t \times F(U^{n+1/2}), \tag{7}$$

where the superscript is devoted to the time step. We use the CFL stability criteria (Courant et al., 1928) to define the maximal stable time-step as :

$$\Delta t \leq 0.5 \frac{\Delta x_{min}}{a} \quad \text{with} \quad a = \max(a_p, -a_m) \tag{8}$$

where $a$ is the magnitude of the velocity of waves, $a_p$ the maximum value of $u_i + \sqrt{gh_j}$ and $a_m$ the minimum value of $u_i - \sqrt{gh_j}$ for $j \in \{i-1; i; i+1\}$ and $\forall i$. $\Delta x_{min}$ is the minimal cell size on the grid.

## 10  2.2  Flux calculation and well-balanced gravity source term on abrupt topography

To compute the numerical flux $F(U^n)$ between two cells, we use the "HLLC" solver, (Toro, 2019) and (Toro et al., 1994), as approximate Riemann solver, which uses a MUSCL-type reconstruction in space with a generalized minmod limiter where $\theta = 1.3$ (*e.g.* (Van Leer, 1979)). This solver conserves the positivity of the water depth and the equilibrium states known as "lake-at-rest" states thanks to the hydrostatic reconstruction of Audusse et al. (2004). Delestre et al. (2012) have shown that

this scheme can non-physically prevent water from flowing when the topography is very steep and the water layer is thin. This situation is far from being trivial since it inevitably occurs as soon as it rains. This is why we add the new second-order reconstruction introduced by (Buttinger-Kreuzhuber et al., 2019; Horváth et al., 2020), and derived from (Chen and Noelle, 2017), which makes it possible to remedy this problem.

### 2.3  Additional source terms

We treat all the other source terms with the time-splitting technique. If we call $S$ the sum of all source terms other than gravity, then the final line of the time-scheme (7) can be written as:

$$U^{n+1} = U^n + \Delta t \times F(U^{n+1/2}) + \Delta t \times S. \tag{9}$$

We will describe in the following paragraphs the different terms that can be modeled in this source term $S$.

### 2.3.1  Rain

The rain is simply treated as: $S_{rain} = R$ where $R$ is the local intensity of the rain in $m.s^{-1}$. This source term is added in the mass equation (1) and was validated in a previous study on Basilisk (Kirstetter et al., 2016). Note that it is possible to integrate rain from Meteo-France Panthere RADAR data directly in the code thanks to the read_lamedeau() function (cf. (METEO-FRANCE, 2020)).





### 2.3.2 Infiltration

We have integrated the infiltration source term of the Green-Ampt model (e-g (Green and Ampt, 1911) ). This model allows us to consider infiltration depending on the hydraulic conductivity $K$, the free space in the porosity material $\Delta\theta$, the wetting front capillary pressure $\psi$ and the volume infiltrated $V$.

$$S_{Inf} = K\left(1 + \frac{\Delta\theta\psi}{V}\right) \tag{10}$$

This source term is also added in the mass equation (1).

### 2.3.3 Friction

Two different friction models are implemented, using the Manning or Darcy-Weisbach relations. These terms are explicitly written as:

$$S_{manning}^{expl} = -n^2 g \frac{q|q|}{h^{7/3}}, \tag{11}$$
$$S_{darcy}^{expl} = -\frac{f}{8} \frac{q|q|}{h^2}, \tag{12}$$

where $n$ is the coefficient of Manning, $f$ the coefficient of Darcy and $g$ the acceleration of gravity.

As recommended in a previous study of (Delestre et al., 2009) we treat the friction term in a semi-implicit way. This leads to
15 the modification of the velocity field obtained from $u = q/h$. The velocity along $x$ is changed as follows for Manning's law :

$$u_x = \frac{u_x}{1 + \Delta t \frac{gn^2|u|}{h^{4/3}}}, \tag{13}$$

and for Darcy-Weisbach's law :

$$u_x = \frac{u_x}{1 + \Delta t \frac{f|u|}{8h}}. \tag{14}$$

The same transformation is applied on the $y$ component of the velocity field.

### 2.3.4 Velocity threshold

When it rains on a very steep topography, waterfalls can occur. In this case, the physics describing the phenomenon is very different from the laws of turbulent friction that we use. It can therefore appear in the simulations speeds that are too high and do not correspond to any physical reality. This is why we have developed a simple velocity transformation that prevents the norm of the vector velocity field from exceeding a certain threshold value set by the user. It is written as:

$$u_x = \frac{u_x T}{|u|}, \text{if } |u| > T \tag{15}$$

where $T$ is the threshold value. The same transformation is applied on the $y$ component of the velocity field.





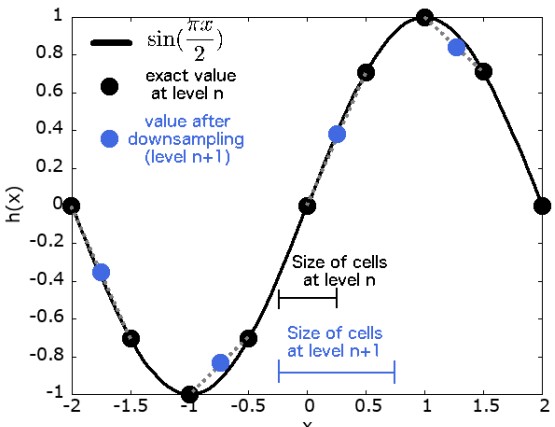

**Figure 1.** Passing from level n to level n+1 (downsampling method)

**Figure 2.** Passing from level n+1 to level n and estimation of the error (upsampling method)

## 2.4 Adaptive Mesh Refinement (AMR)

The b-flood software takes advantage, among others, of the adaptive mesh refinement (AMR) technique developed on Basilisk by S.Popinet (Popinet, 2015). This process is very well explained in van Hooft et al. (2018) as well as in the sandbox of Van Hooft on the Basilisk website (van Hooft, 2020). We recall here the general mechanism, drawing heavily on the previously cited publications. The `adapt_wavelet()` function allows to refine or to enlarge the mesh according to the error estimated by the algorithm between two levels of refinement on one or more scalar or vector fields. The error criterion and the fields concerned are set by the user. If the error estimated is greater than the user-defined criterion, the algorithm will refine the cell into four smaller daughter cells. If the error estimate is less than two thirds of the same criterion, then the algorithm considers that the resolution is too fine and thus that the computing time is increased. The function will therefore combine the 4 cells concerned into a single large cell. If the error is between two thirds of the criterion and once the criterion, then nothing happens. The criterion can finally be seen as the maximum permissible error between two levels of refinement. The process of the evaluation of the error on the example of a perfect sinusoidal swell is described on the figures 1 and 2. It is important to note that the function used to calculate the new field value after refining or "coarsening" the cells is usually a linear interpolation, but it can be set by the user differently for each field. For example, the function used to refine the topography is not a linear interpolation, but simply the value that the topography had before coarsening. The practical use of this function is detailed in the 3 examples of real cases published below. In addition to the adaptive refinement process, it is possible to refine certain parts of the mesh statically using the refine() and coarsen() functions.





## 2.5 b-flood: a subset of Basilisk

In practice, the b-flood software is a sub-part of the open-source Basilisk software, created by S. Popinet (Popinet, 2013). This means that it takes advantage of all the features of Basilisk: it is completely free and open-source. In addition to the AMR described above, it also allows parallel computing. Experienced users will be able to develop their own modules while simple

users will be able to copy/paste the sample scripts given in the rest of this article. It is still recommended to learn how to use the basics of Basilisk before using b-flood, thanks to the various tutorials available on the website (basilisk.fr).

## 3 Evaluation of the performance on test cases

### 3.1 Analytical test cases

It is important to ensure that the numerical schemes we use are consistent. We test them on complete benchmarks, i.e. the

transition from subcritical flow, when the wave velocity is higher than the flow velocity, to supercritical flow, as well as the transition from supercritical to subcritical flow which is characterized by the presence of a shock. The following test cases can be found in the software SWASHES published in (Delestre et al., 2013). They are designed to test the validity of implementation of source terms and the consistency of the numerical scheme by comparing the numerical solution with the analytical one.

### 3.1.1 Subcritical to supercritical flow with Manning friction

In this benchmark, we test the transition from a subcritical to a supercritical regime with Manning's law of friction. A constant flow $q_0 = 2 \ m^2.s^{-1}$ is imposed on the left boundary of the flume on the topography plotted in Figure 3 . The case is in 1D and the domain is 1000 meters long and initially dry. The friction is modeled by the Manning law with $n = 0.0218 m^{-1/3}.s$. The flow is subcritical for the coordinate $x < 500 \ m$ and supercritical otherwise. The duration of the experiment is $2000s$. We checked that the flow is stationary at the end of the run. At the end of the experiment, we compute the following error norms :

$$
\begin{aligned}
n1 &= \frac{\Sigma_i |h_i - he_i|}{N}, \\
n2 &= \sqrt{\frac{\Sigma_i (h_i - he_i)^2}{N}}
\end{aligned}
\tag{16}
$$

with $N$ the total number of cells, $h_i$ the water depth computed on b-flood at the cell $i$, $he_i$ the exact solution for the water depth at the location of the cell $i$ and $\Sigma_i$ the operator to sum over all the cells.

The water heights profile at resolutions $N = 32$ and $N = 512$ are compared to the exact solution on figure 3. The convergence

of the different error norms according to the resolution is shown on figure 4 in logscale for both resolution axis. As we can see, the simulated profile converges toward the analytical profile and all the error norms converge to zero with an order larger than one.





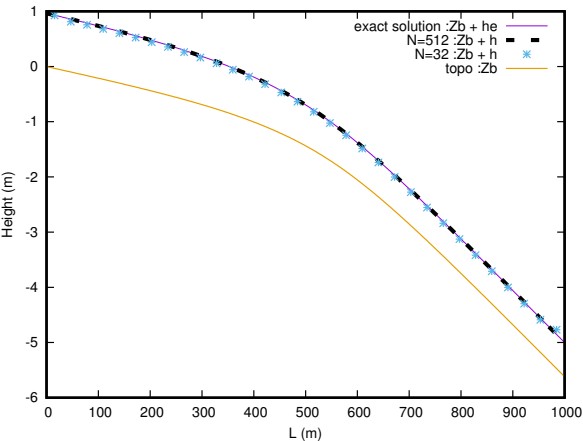

**Figure 3.** Comparison of the water heights profiles between the simulation with the analytical solution of the Manning friction test case for two different resolutions.

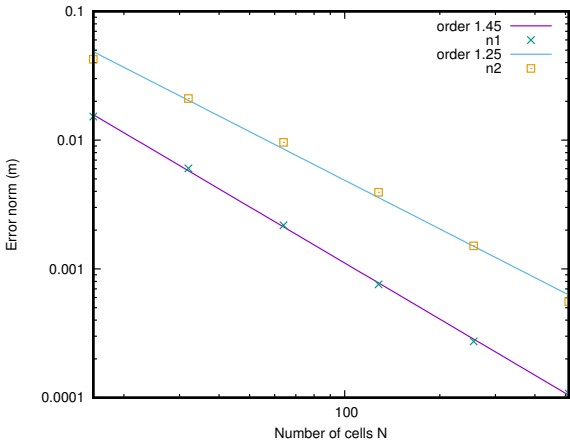

**Figure 4.** Convergence of the different error norm when resolution is increasing.

### 3.1.2 Transonic transition and shock with Darcy-Weisbach friction

In this benchmark, we validate the transition from a supercritical flow to a subcritical flow which is characterized by the presence of a shock. The domain is 100 meters long and a constant discharge of $q_0 = 2\ m^2.s^{-1}$ is imposed on the left boundary (upstream) on the topography plotted on the figure 5. At the right boundary (downstream), the water height is fixed

to its analytical value. The flow is subcritical at the left of the slope, becomes supercritical via a sonic point and then becomes subcritical again via a shock. The case is in 1D and the friction is modeled by the Darcy-Weisbach's law with a coefficient $f = 0.093$. The duration of the experiment is $200\ s$. We checked that the flow is stationary at the end of the run.

As done before, the water depth profile at resolutions $N = 32$ and $N = 512$ are compared to the exact solution on figure 5. In the same figure, we represent the distribution of the error: $h - he$ as a function of the position. We can see that a large error

is found at the shock location. This is due to the fact that the position of the shock is necessarily approached at the resolution step $dx$. The convergence of the different error norms according to the resolution can be seen on figure 6. The presence of the error on the location of the shock necessarily induces orders of convergence smaller than in the previous case, but sufficiently convincing for the convergence of the code.

### 3.2 b-flood versus experimentation: "the Toce model"

In order to validate b-flood, we use a flood experiment recreated by researchers: the Toce Valley model. The Toce model was designed to study the capabilities of different numerical models to accurately represent the flow characteristics. This model was done for the CADAM project by Soares et al. (Soares Frazao and Testa, 1999). It is a reproduction of the Toce river valley





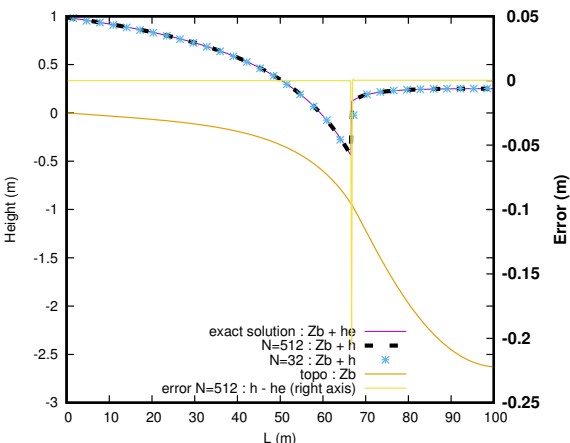

**Figure 5.** Comparison between the simulation with the analytical solution of the transonic test case for two different resolutions.

**Figure 6.** Convergence of the different error norms when the resolution is increasing.

in Italy at the scale $1 : 100$. It is fully instrumented with many stations that measure water level profiles at multiple locations. A nozzle is placed at the entrance of the domain to deliver a controlled flow rate.

### 3.2.1 Fluvial case

The first case is done on the entire river which is 50 meters long and 11 meters large. The DTM is at a resolution of 5 cm, note that this includes the reproduction of houses. The topography with the position of the 21 gauge stations can be seen on figure 7, where the downward direction is from left to right. The missing numbers are gauges that did not work during the experiment and therefore were not provided by the experimenters. The imposed inlet condition is an hydrograph. The hydrograph is composed of a brutal rising stage from 0 to $210 \, l.s^{-1}$ then by a slower and continuous descent phase which reaches up to $60 \, l.s^{-1}$ at the end of the experiment, as we can see in figure 8. The duration of the experiment is $180 \, s$.

We reproduce the exact same case with b-flood with a minimal cell size of $\Delta_{min} = 4.2 \, cm$ and a maximal one of $\Delta_{max} = 67.8 \, cm$. We model the friction with Manning's law and we set Manning's coefficient to $n = 0.0162 \, m^{-1/3} \cdot s$ as recommended by the CADAM report. On the left edge, we impose as an inlet condition a water elevation on the edge such that the flow is that imposed by the hydrograph. The boundary condition on the normal velocity is a Neumann condition: $\partial_x U_x = 0$ and on the tangential velocity, a Dirichlet condition: $U_y = 0$. We make sure that we impose the right flow rate by comparing the volume in the b-flood simulation with the imposed flow rate, converted to volume using the following equation: $Volume(t) = \int_0^t Q_{imp}(t')dt'$. We can see on figure 8 that the correct inflow is imposed in the simulation. Note that the volume in the simulation "stalls" from the imposed volume around $t = 80$ seconds, when the water flow exits the simulation domain by the right edge. On the right edge of the domain, we set a condition of free exit of water and flow. For adaptive refinement, the error threshold is set at $5 \, mm$ on the water level field. To ensure the exact reproduction of the experimental case, some precautions

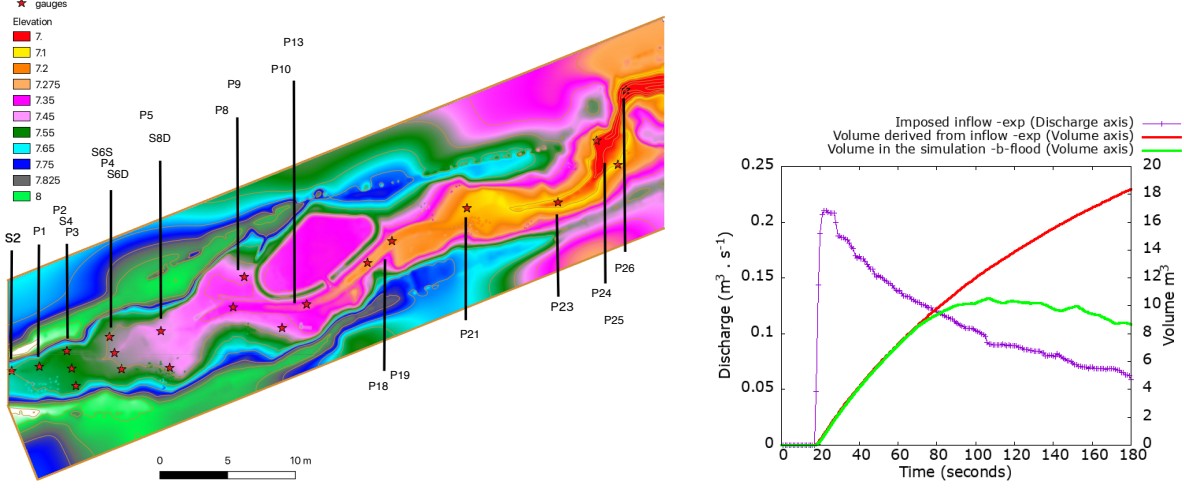

**Figure 7.** Topography of the fluvial case with position and name of the gauge stations. The inflow is coming from the left border of the domain.

**Figure 8.** Hydrograph of the imposed inflow on the left boundary.

must be taken. We artificially set a small time step $\Delta t = 0.01\ s$ at time $t = 17\ s$ to make sure we capture the short rise of the hydrograph. Still with the aim of capturing the rise of the hydrograph, we leave off the adaptive refinement until time $t = 18\ s$. Note that we do this because the simulation domain is empty for the first 17 seconds, which will not happen in a real case where rivers are flowing and rain is falling. The simulation runs in 18966 seconds on an Apple laptop equipped with a 2.8 GHz Intel

Core i5 dual-core processor. We measure the water depth profiles at the exact positions of the 22 gauge stations and we record movies of the flow characteristics during the experiment: water depth, velocity of flow and Froude number. All the movies are available on the b-flood website, as are other data and the entire code. We can see the flood wave front propagation as well as the resulting automatic adaptive refinement in figure 9.

    To quantify the performance of b-flood, we define the following different error standards, starting with $e_1$ (meters) and

$e_{1r}$ (no unit):

$$e_1 = \frac{1}{t_{end} - t_s} \int_{t_s}^{t_{end}} (h_{num}(t) - h_{exp}(t))\ dt \tag{17}$$

$$e_{1r} = \frac{\int_{t_s}^{t_{end}} (h_{num}(t) - h_{exp}(t))\ dt}{\int_{t_s}^{t_{end}} h_{exp}(t)\ dt} \tag{18}$$

where $t_{end}$ is the final time of the experiment, $h_{exp}$ is the experimental value of the water depth at the considered water gauge, $h_{num}$ is the value of the water depth found by b-flood at the same location. The time $t_s$ is defined as the time when both

numerical and experimental values of the water depth exceed the threshold value of $5\ mm$. Note that $e_1$ and $e_{1r}$ are positive if $h_{num}$ is mostly greater than $h_{exp}$, and vice versa. $e_{1r}$ is equal to $e_1$ normalized by the mean height of the experimental case. $e_{1r}$ should be read as the mean percentage error with respect to the experimental height.



**Figure 9.** Picture of the water depth (A) and the refinement level (B) at $t = 21\ s$ (a), $t = 35\ s$ (b), $t = 49\ s$ (c) and $t = 63\ s$ (d)





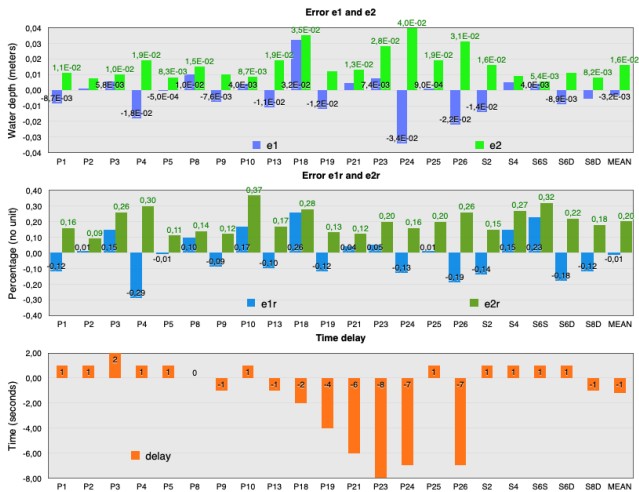

Figure 10. Error norms ant time delay with respect to the gauge station.

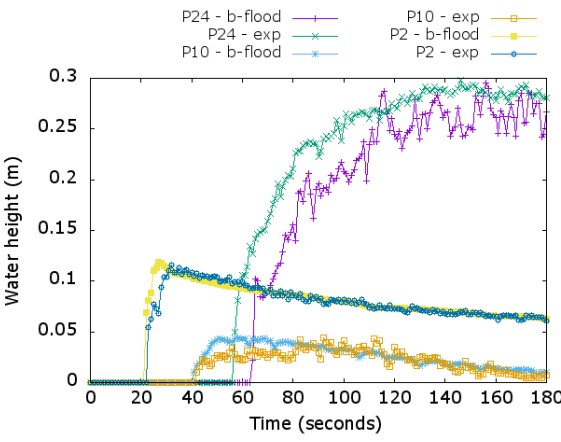

Figure 11. Water depth for experiment and b-flood for gauge stations P2, P10 and P24.

We define also $e_2$ and $e_{2r}$ as :

$$e_2 = \sqrt{\frac{1}{t_{end} - t_s} \int\limits_{t_s}^{t_{end}} (h_{num}(t) - h_{exp}(t))^2 \, dt} \qquad (19)$$

$$e_{2r} = \frac{\sqrt{(t_{end} - t_s) \int_{t_s}^{t_{end}} (h_{num}(t) - h_{exp}(t))^2} \, dt}{\int_{t_s}^{t_{end}} h_{exp}(t) \, dt} \qquad (20)$$

Note that unlike $e_1$ and $e_{1r}$, $e_2$ and $e_{2r}$ are always positive. As $e_{1r}$, $e_{2r}$ is equal to $e_2$ normalized by the mean value of $h_{exp}$.

Also as $e_{1r}$, $e_{2r}$ should be read as the mean percentage of the RMSE with respect to the experimental height.

Finally, we define the delay time as the time between the two instants when at least 5 mm of water arrives at the measuring station, in the real case and in the case simulated on b-flood. This delay time is a good metric to quantify the capacity of b-flood to mimics the dynamics of the experimental case. Note that a positive delay corresponds to the case where water arrives first in the numerical case: delay is positive when b-food is early and delay is negative when b-flood is late.

We report on figure 10 the values of the different error norms. We can see that the mean value of the root mean square error ($e_2$) on all the stations is around 16 $cm$ and is around of 20 % for the root mean square of the relative error ($e_{2r}$). We report on figure 11 the water depth measured at stations P24, P10 and P2. Measuring station 24 has the worst root mean square error and one of the worst time delays. We can see that, although it is slightly underestimating the water level, b-flood does capture the dynamics of the flood wave on this station. Station 10 is the worst in terms of relative error. However, we can see that b-flood

models the flow with a sufficient precision at this station as well. On the other hand, b-flood provides a very good estimate of the flow on station 2.

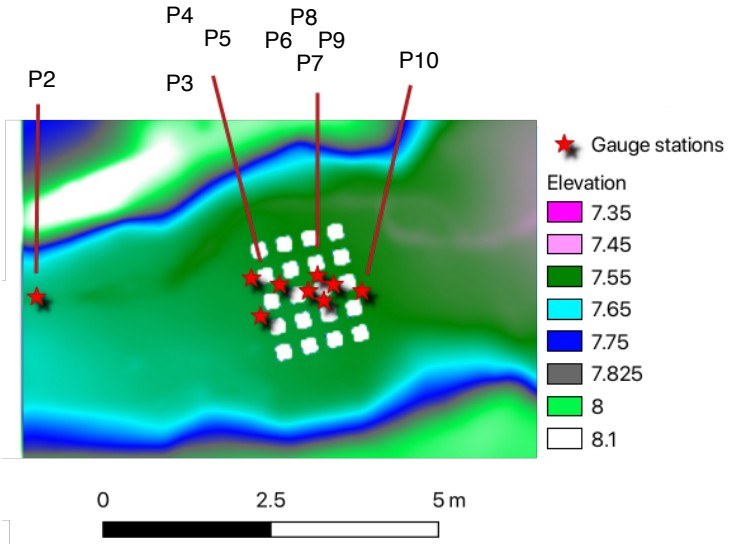

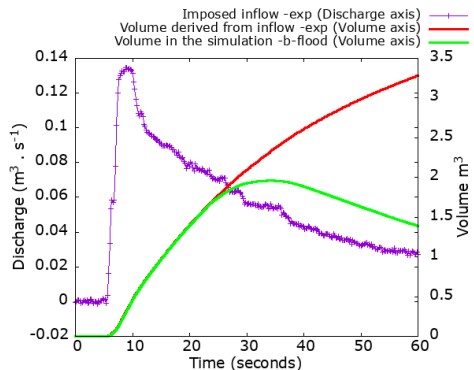

**Figure 12.** Zoom on the first $10\ m$ of the topography of the urban model with positions of buildings and gauge stations.

**Figure 13.** Hydrograph of the imposed inflow on the left boundary

In conclusion, based on these results, we can say that b-flood correctly models fluvial cases of floods on impermeable soil with imposed inflow and the presence of houses.

### 3.2.2 Urban case

The second case of validation is reproducing "The Model city flooding experiment benchmark" presented in (Alcrudo et al.,
2003) and (Testa et al., 2007) (data are freely available from https://www.tandfonline.com/doi/abs/10.1080/00221686.2007.
9521831). This model is build on the first $15\ m$ of the precedent Toce model. The authors added 20 buildings distributed in 4
aligned rows, 9 gauge stations are distributed around the buildings and at the entrance of the flow. The topography of the case
and the location of gauge stations are shown on figure 12. The imposed entry condition is again a hydrograph. The flow rate
goes from 0 to a maximum of $130\ l.s^{-1}$ in 4 seconds and then progressively decreases to $30\ l.s^{-1}$ in 50 seconds, as we can
see on figure 13. This condition reproduces in fact the typical water flow of a flash flood. The experiment lasts 60 seconds.

We reproduce this case using b-flood. The domain consists of cells of size between $\Delta_{\min} = 1.46\ cm$ and $\Delta_{\max} = 23.4\ cm$.
For adaptive refinement, the error threshold is set at $1\ mm$ on the water level field. We set as an entry condition on the left
boundary a constant water height such that the inflow is the one imposed by the hydrograph. The boundary condition on
the normal velocity is a Neumann condition: $\partial_x U_x = 0$ and on the tangential velocity, a Dirichlet condition: $U_y = 0$. Exactly
as done previously for the case validated for the "fluvial Toce". We compare the volume of water entering the simulation
and the volume of water entering the experiment (thanks to the hydrograph) on figure 13, a perfect match is obtained. We
use Manning's friction law with a value: $n = 0.0162\ m^{-1/3}.s$, as recommended by the authors of (Alcrudo et al., 2003). To





replicate the vertical walls of the houses, we raise the topography of our simulation by an immense height at the site. This has the effect of producing near-vertical walls. The simulation runs in 23047 seconds on an Apple laptop equipped with a 2.8 GHz Intel Core i5 dual-core processor. We can see the arrival of the flood wave simulated by b-flood as well as the adaptive refinement on the figure 14. In these figure, we can see that the front of the flood wave is refined to the maximum, but that the

refinement becomes coarse again once the wave has passed, if the water flow is not too complex.

We record in the simulation the water heights at the exact locations where the measurement stations are in the experiment. Then, for each of these stations, we calculate the norms $e_1$, $e_{1r}$, $e_2$ and $e_{2r}$ . We also calculate the delay time between the arrival of the flood wave in the numerical case and in the experimental case. These results are shown on the figure 15. We can see that the absolute mean error remains more or less the same as in the fluvial case with a mean value of $1.9 \ cm$. However,

the relative error is greater than the previous case with a mean of $42\%$. This value may seem high, but it mainly reflects the error made on station 5. In figure 16, the water height recorded at station P5 is shown, which gives the worst result. It should be noted that other studies on this case also give "bad" results on this station and not on the others. As an example, we have given the results of (Kim et al., 2014) that the authors obtained on the exact same case with their Saint-Venant solver on an unstructured grid using triangular elements. This leads us to believe that the error comes from the presence of a hydraulic jump

that the Saint-Venant equations do not allow to predict correctly, and therefore cannot be attributed to the numerical method. The delay time does not exceed 5 seconds for all the stations and its maximum value is on station P2. We can see in figure 16 that the dynamics modeling therefore remains convincing.

The results produced allow us to conclude on the validity of our simulations in the case of urban flooding on impermeable soil.

**4  Real case: flood of October 2015 in Cannes on the French Riviera**

Here we demonstrate the possibility of using b-flood, a software based on shallow water equations and mesh refinement, in a real situation of flash flood on small watersheds (less than $100 \text{km}^2$). For this, we simulate the case of the flash flood that took place in Cannes (France) on October 3, 2015 (Carrega, 2016; Saint-Martin et al., 2018). The city of Cannes is located in the south-east of France. The Saturday third of October 2015, between 6 pm and 11 pm, a huge rain fell on the Alpes-Maritimes

department in France: at some points, $200 \ mm$ of rain in less then 3 hours were recorded. This outstanding meteorological event killed 20 people and the CCR ("Caisse Centrale de Réassurance" – www.ccr.fr which is a French public re-insurer) estimated the total material lost between 500 M¤ and 650 M¤. Around the river Siagne, the SISA ("Syndicat Intercommunal de la Siagne et ses Affluents" which aimed at fighting against floods on the territory of the member municipalities. Today, SISA has been dissolved, and the missions have been taken over by the SMIAGE since 1 January 2018.) recorded a rain intensity

larger than $200 \ mm$ per hour around $8:30$ pm. This meteorological event, where a large amount of rain is localized on a small area during a short laps of time, is known as a "flash flood", and has caused the appearance of torrents of water throughout the streets of Cannes. Moreover, in the upstream part, the soil was already saturated by a heavy rain which occurred on October 2 and in the city the storm-water system was also saturated.

**Figure 14.** Picture of the water depth (A) and the refinement level (B) at $t = 10\ s$ (a), $t = 13\ s$ (b), $t = 17\ s$ (c) and $t = 26\ s$ (d)





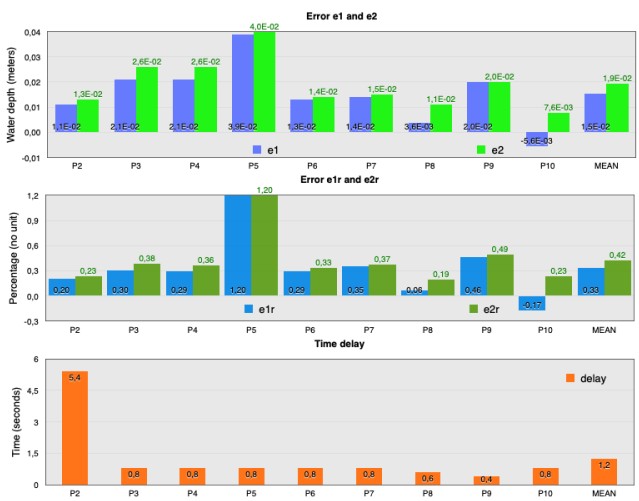

**Figure 15.** Error norms with respect to the gauge stations and their mean values.

**Figure 16.** Water depth for experiment and b-flood for gauge stations P5 and P2.

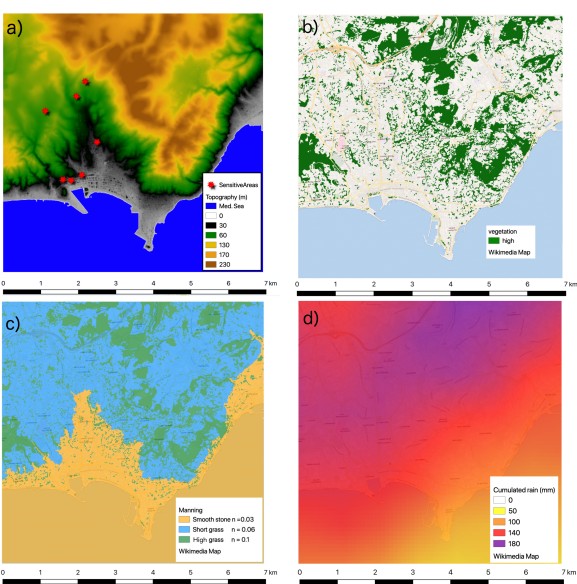

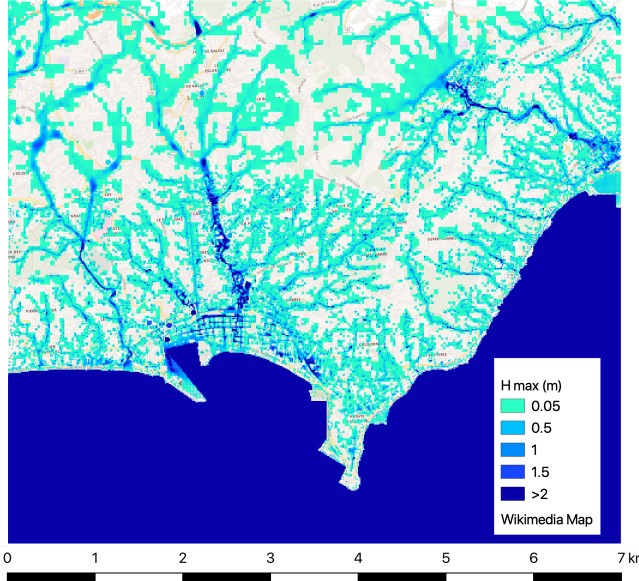

**Figure 17.** a) Topography of the simulated domain and position of sensitive areas. b) High vegetation zone. c) Manning coefficient values. d) Accumulated rain during the rain event.

**Figure 18.** Flood extent of the event simulated by b-flood.



To simulate this event with b-flood, we use a Digital Terrain Model at a resolution of 1 m, courtesy of the IGN (RGE-ALTI). We also use the Digital Surface Model to add buildings. The buildings are simulated thanks to an elevation of the topography corresponding to their real heights. The total size of the domain is $7\ km\ \times\ 7\ km$ and fully encompasses the catchment area of the city of Cannes. The maximum cell size is $\Delta x_{\max} = 235\ m$ and the minimum size is $\Delta x_{\min} = 13.6\ m$. We use adaptive

refinement with a threshold value of $5\ cm$ on the height of water. We added an even smaller cell size: $\Delta x_{spe} = 6.8\ m$ to mesh more precisely specific sensitive areas: town halls, fire stations, hospitals and police stations. This mesh size allows us to be more accurate on these sensitive areas without slowing down our simulation. The DTM + DSM topography used as well as the location of sensitive areas can be seen in figure 17 -a).

We use Manning's law and the infiltration source term. IGN also provides soil plant occupation maps (BD TOPO). We can

see on the figure 17-b) the zones of high vegetation. We use these areas to set the value of the Manning coefficient as well as the various infiltration parameters. The Manning coefficient is set to $n = 0.1\ m^{-1/3}\cdot s$ in high vegetation zones, $n = 0.03\ m^{-1/3}\cdot s$ where the topography is below 50 m to represent the urban zone and is set to $n = 0.06\ m^{-1/3}\cdot s$ everywhere else, see figure 17-c). The infiltration parameters are set to loamy sand values on high vegetation areas ($\psi = 6.1\ cm$, $K = 3e-5\ cm\cdot h^{1}$, $\theta = 5\%$) and to values of sandy clay everywhere else ($\psi = 21\ cm$, $K = 0.15e-5\ cm\cdot h^{1}$, $\theta = 5\%$).

We use the source term of rain to add the precipitations measured by Meteo-France and which are provided free of charge (RADAR PANTHERE). These data are at 5 minutes time steps and the pixels are of size $1km \times 1km$. We can see the cumulated rain during the event on figure 17-d).

We fix the threshold value on the celerity to $V_{threshold} = 10\ m.s^{-1}$ in order to avoid slowing down the code unnecessarily. In fact, when rain is added to a quasi-vertical topography, the speed of the water can reach high values which, in addition to

not representing reality, slows down the code. The duration of the simulated event is 5 hours. The simulation was performed on a 16 cores desktop in 6472 seconds, which is consistent with the time delay of rainfall predictions. We record the maximal value of the water depth field during the entire event, as shown on figure 18 highlighting the flood extent. Movies of evolution of the water depth, the celerity and of the level of refinement can be seen online on the website of b-flood (Kirstetter, Delestre, Lagrée, Popinet and Josserand, 2019), showing a good qualitative behavior of b-flood to reproduce the whole event.

## 5 Conclusions

This paper presented b-flood, an open-source Saint-Venant model for simulations of surface flows in two dimensions using adaptive refinement. The code is completely free and open-source, like the Basilisk software from which it is derived. The model uses a well-balanced scheme that does not prevent water from flowing over steep topography.

The validity of the numerical scheme has been tested on two analytical benchmarks. The convergence of the scheme has

been observed with a good order of convergence. The code has been then tested on two experimental cases in the Toce Valley, one fluvial and the other urban. The results of the simulation gave satisfactory agreement with the experimental results. Finally, we demonstrated the practical effectiveness of b-flood on a real case of flash-flood on a small watershed in south of France: the October 2015 flooding of the city of Cannes, French Riviera. This event has caused 20 fatalities and a lot of material





damage. The city of Cannes has faced 200 mm precipitation in less than 3 hours. In the upstream part, the soil was already saturated by a heavy rain which occurred on the 2nd of October and in the city the storm-water system was also saturated. This has demonstrated the feasibility of using a software based on shallow water equations and mesh refinement for flash flood simulation on small watersheds (less than $100km^2$). Remarkably, for this practical case, predictions of the flood dynamics and

5   localization could be deduced in a computational time compatible with the rainfall predictions, opening the way to real time flood forecasting.

Future work will focus on (1) implementing hydraulic structures such as culverts, gates, weirs and (2) coupling this overland flow model with a storm-water network model. This will improve b-flood's capability of performing more complete flash flood simulations, particularly in south of France watersheds.

*Code and data availability.*   You will find the code (Open Source) as well as the different data files to be able to reproduce our results here : http://basilisk.fr/sandbox/B-flood/Readme. It should be noted that we do not have the right to distribute the DTM file of the IGN's topography of Cannes as well as the METEOFRANCE files. The case of Cannes is therefore unfortunately not reproducible.

20   *Competing interests.*   We declare that no competing interests are present



*Acknowledgements.* We would like to thank F. Bourgin for the initial idea of this paper. We would like to thank the IGN for providing us with the topographic data as well as Météo-France for providing us with the data of their RADAR PANTHERE. We also thank the AXA Foundation for Research fund for their financial support at the very beginning of this study.





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
