# Peer review of "b-flood 1.0: an open-source Saint-Venant model for flash flood simulation using adaptive refinement"

_Geoscientific Model Development, 2021_

## Referee Comment (RC1)

Report on *B-flood 1.0: an open-source Saint-Venant model for flash flood simulation using adaptative refinement*,
by G. Kirstetter, O. Delestre, P.-Y. Lagrée, S. Popinet, C. Josserand

This article presents the b-flood software, subset of the Basilisk library, that solves the Shallow-Water equations with an adaptative mesh refinement. The software is tested against analytical solutions and is confronted to experimental datas in cases of flash flood.

**General comments**

The b-flood software solves the Shallow-Water equations with a finite volume approach, on a square grid. The HLLC solver and a MUSCL-type reconstruction are implemented, as well as the hydrostatic reconstruction to preserve the positivity and equilibrium states. Two points must be underlined:

- thanks to a second order reconstruction, the software can deal with very steep topographies,

- a threshold is introduces on the velocity to avoid large values due to some buildings. These large values could generate very small time steps.

These modifications are key tools to be able to simulate urban cases under flash flood.

After describing the software, the authors give two examples of classical analytical solutions that are well reproduced by b-flood. Convergence orders are shown. Then two experimentations on the Toce model are simulated: a reproduction of the Toce river valley (Italy) was built at the scale 1:100 and several experiments were performed. The authors chose a fluvial case and a urban, torrential case, and they confront their results to the measurements: there is a good fit. The last part of the article is devoted to the simulation of a real case, the flood of October 2015 in Cannes. They obtain a good qualitative behavior of the software.

**I suggest to accept the publication of this article after minor corrections and answers to the following questions.**

**Specific comments**

- section 2.3.4 velocity threshold: could you justify the threshold ? Do you have a reference ? Could you explain more this choice ?

- section 2.4 AMR line 8: why do you choose 2/3 ? Is it usual / your choice ?

- Fig 8 and 13: To avoid the difference between the experimental and b-flood results, why don't you add the outlet volume of water ? It should then fit the red curve ?

- Fig 10 and 15: what is the use of the time delay ? You did not comment on these results.

- page 13 lines 1-2, and page 14 lines 18-19: One can say that b-flood is able to reproduce this Toce example, it cannot be generalized to all the floods on impermeable soil with houses.

- page 14 line 22: could you explain the limit to small watersheds (less than 100 km$^2$)?

- page 17 lines 1-8: is it usefull to have a DMT with a resolution of 1 m when the space step is of several meters ?
  How do you consider a river that is smaller than the space step ? How can the flow be continuous if the river is "crossing" the mesh ?

- Fig 18: you claim a "good qualitative behavior" but no measurements are shown to compare the simulated results and what happened.
  Could you explain the differences with the results of:
  G. Kirstetter, F. Bourgin, P. Brigode, O. Delestre. *Real-Time Inundation Mapping with a 2D Hydraulic Modelling Tool Based on Adaptive Grid Refinement: The Case of the October 2015 French Riviera Flood.* Advances in Hydroinformatics, pp. 335-346, 2020 (Simhydro2019), obtained with Basilisk ?
  This reference should be added and explained in the article.

**Technical corrections**

- Is there a capital letter in the name of the software ? Homogenize the title and the rest of the paper.

- page 1 line 13: in the Gard region, the flood occurred in September, not June 2002.

- page 3 line 24: () are forgotten in the equation number.

- page 4 line 9 : $\forall i.\Delta x_{min}$ could be misunderstood, add a word at the beginning of the last sentence.

- page 9 line 6 : I don't understand the "missing numbers", each gauge has a number ... ?

- Fig 7: Impossible to read the legend when the article is printed, enlarge the figure ?
  What are the meanings of P, S ?

- Fig 8: the volume is not mentioned in the caption.

- Fig 9: no legend: values of the water height and of the mesh refinement ?

- Fig 10: Impossible to read the values when the article is printed, enlarge the figure ?
  Typo : "ant" in the caption.

- page 12 line 6: "time delay" instead of "delay time".

- Fig 12 : the names of the gauges are the same as fig 7 but they are not located at the same places.

- Fig 13: the volume is not mentioned in the caption.

- page 13 line 15: sentence which does not contain a verb.

- page 14 line 1: refer to section 2.2 for the abrupt topography.

- page 14 line 4: a "s" is missing in "figures" or change "these".

- page 14 line 27: the euros (dollars ?) symbol must be changed twice.

- Fig 14: no legend: values of the water height and of the mesh refinement ?

- Fig 15: Impossible to read the values when the article is printed, enlarge the figure ?
  The time delay is not mentioned in the caption.

- Fig 17: Impossible to read the legend when the article is printed, enlarge the figure ?

- page 17 line 18 : refer to section 2.3.4 for the threshold.

- page 22 line 5 : "B-flood" to change into "b-flood" ?
  The link is not correct, one must change the B to the lower case.
  Also add the first names of the authors.

---

## Referee Comment (RC2)

**Paper :** B-flood 1.0: an open-source Saint-Venant model for flash flood simulation using adaptive refinement

**Authors : Geoffroy Kirstetter, Olivier Delestre, Pierre-Yves Lagrée, Stéphane Popinet, and Christophe Josserand**

**Review :** The paper deals with the numerical simulation of flash floods using a numerical software based on the two dimensional shallow water equations. The paper is well presented. The equations, the numerical scheme and the adaptative mesh refinement strategy are presented in Section 2. In Section 3, the outputs of the software are compared first to analytical solutions and then to laboratory experiments. Eventually, in the last section of the paper, the authors reproduce a flash food event that occured in south-east France in October 2015. It results from this study that the software is able to give reasonably accurate results in a CPU time that is compatible with online use. It seems to me that the paper could be accepted for publication in GMD wtih minor modifications.

Comments and questions :

- On the CFL condition (Section 2.1, equation 8 in the manuscript) :

    - First, I'm not sure that the velocity $u = q/h$ has been introduced before.
    - Why do the authors not use the less restrictive CFL condition

    $$\Delta t \leq 0,5 \min_i \frac{(\Delta x)_i}{a_i} \ ?$$

    - What motivates the choice

    $$(a_p)_i = \max_{j=i-1,i,i+1} (u_i + gh_j),$$

    instead of the classical choice $(a_p)_i = u_i + gh_i$ ?
    - What is the form of the CFL condition for the 2d computations ?

- On the computation of the source term (Section 2.2 in the manuscript) :

    - The authors propose a second order accurate (in time and space) discretisation of the flux term but a first order in time discretization of the source term. Would it be possible to propose a stable second order in time discretisation of the source term, especially for the friction term, and then to obtain global second order accuracy in Figure 4 ?
    - Rain and infiltration source terms are added in the mass equation only. How do the authors justify that they have no impact on the momentum equation ?

– Equations (13) and (15) should be rewritten. The equality is valid in a code where we assign a value but not in a mathematical sense.

- On the comparison with analytical solutions and laboratory experiments (Section 3 in the manuscript) :

    – Since it happens in flash flood events, it would be interesting to test the software agains an analyical solution with dry areas, as the well known Thacker test case.

    – I do not understand what the sentence "For adaptive refinement, the error threshold is set at 5 mm on the water level field" means. Doesit mean that there is only two level of refinement, one for value of $h$ lower than the threshold and one for value of $h$ greater than the threshold ? It does not seem to be the case on the numerical results, Figure 9.

    – "Errors" defined by (17)-(18) are not errors since they can vanish for nonidentical solutions. The only errors here are the L2 errors defined by (19)-(20). Another word has to be used.

    – The words "and the presence of houses" at the end of Section 3.2.1 sshould be removed since the houses are introduced in the next section.

- On the flash flood simulation (Section 4 in the manuscript) :

    – It would be interesting to give some snapshots of the meh at different times to see the effect of the AMR strategy.

    – "We record the maximal value of the water depth field during the entire event, as shown on figure 18 highlighting the flood extent." Is it possible to have acces to a map of the flood as it happens in October 2015 or at least to some data at certain location that could be used for comparison with the numerical results ?

    – I know it is a huge work (probably out of the scope of this paper), but it would be interesting to analyse the sensibility of the results to the friction and infiltration coefficients. In particular, a first attempt could be to reproduce the same event but with a larger high vegetation zone and to see if the results are significantly different or not.

---

## Author Comment (AC1)

We would like to thank the referee. These pertinent remarks allowed us to improve our article. We answer here point by point to these remarks and we modified the article when it was necessary.

- *section 2.3.4 velocity threshold: could you justify the threshold ? Do you have a reference ? Could you explain more this choice ?*

    This threshold can be explained physically by the real friction wich occurs on flows. The friction models used can miss this physical threshold in rare cases (quasi-vertical slope with rain for example). This results in the appearance of abnormally high speeds in small areas of simulation. These velocities do not question the validity of the simulation because it concerns very small quantities of water, but on the other hand it drastically decreases the time step and thus increases significantly the simulation time. The velocity threshold is chosen in the order of 10 m/s in an empirical way. We verify at the end of the simulation that this regularization of the velocity has been done in very small areas not questioning the validity of the simulation.

- *section 2.4 AMR line 8: why do you choose 2/3 ? Is it usual / your choice ?*

    The value must be less than 1 to create a hysteresis cycle and thus avoid the  refinement / unrefinement round trips in loop. This is a classic value.

- *Fig 8 and 13: To avoid the difference between the experimental and b-flood results, why don't you add the outlet volume of water ? It should then fit the red curve ?*

    Yes, by adding the volume of water leaving the simulation, the curves would overlap. This volume must be calculated and is not a direct data. In order not to add complications, and verifications, we have chosen to represent only data directly extracted from our simulation without any calculation on our part. This is enough to prove that our inflow is the right one.

- *Fig 10 and 15: what is the use of the time delay ? You did not comment on these results.*

    In order to avoid any confusion and thanks to the referee's feedback, we have changed the name of this time to "arrival time delay". The definition of this term can be found on p.12 l.8 of the article. We have also added comments on this term in the "Fluvial case" and "Urban case" sections.

- *page 13 lines 1-2, and page 14 lines 18-19: One can say that b-flood is able to reproduce this Toce example, it cannot be generalized to all the floods on impermeable soil with houses.*

    Indeed, we have overgeneralized this case study. We have modified the article accordingly.

- *page 14 line 22: could you explain the limit to small watersheds (less than 100 km2 )?*

    This is a usual limit. This roughly determines the responding watersheds with characteristic times of the order of an hour. They are thus particularly well adapted to this type of simulation.

- *page 17 lines 1-8: is it usefull to have a DMT with a resolution of 1 m when the space step is of several meters ?*

    The DMT used is the one provided by the IGN. The code then averages it to make a coarse grid. The more accurate the grid used, the more accurate the coarse grids will be.

- *How do you consider a river that is smaller than the space step ?*

A river smaller than the space step will therefore influence the coarse grid: the reconstructed coarse value will therefore be lower.

- *How can the flow be continuous if the river is "crossing" the mesh ?*

I don't quite understand the question. The water flow is always discontinuous and this does not hinder its modeling: it is a property of the finite volume method.

- *Fig 18: you claim a "good qualitative behavior" but no measurements are shown to compare the simulated results and what happened. Could you explain the differences with the results of: G. Kirstetter, F. Bourgin, P. Brigode, O. Delestre. Real-Time Inundation Mapping with a 2D Hydraulic Modelling Tool Based on Adaptive Grid Refinement: The Case of the October 2015 French Riviera Flood. Advances in Hydroinformatics, pp. 335-346, 2020 (Simhydro2019), obtained with Basilisk ? This reference should be added and explained in the article.*

That's an excellent point. This other article was done later but was published earlier, which is why we forgot to mention it here. We have added a paragraph in the article about it.

**Technical corrections**
- *Is there a capital letter in the name of the software ? Homogenize the title and the rest of the paper.*
  Done.
- *page 1 line 13: in the Gard region, the flood occurred in September, not June 2002.*
  Done.
- *page 3 line 24: () are forgotten in the equation number.*
  Done
- *page 4 line 9 : $\forall i. \Delta x min$ could be misunderstood, add a word at the beginning of the last sentence.*
  This paragraph has been corrected
- *page 9 line 6 : I don't understand the "missing numbers", each gauge has a number ... ?*
  Explanation added in the article.
- *Fig 7: Impossible to read the legend when the article is printed, enlarge the figure ?*
  Done.
- *What are the meanings of P, S ?*
  These are the two types of measuring stations. You can find more information in their articles.
- *Fig 8: the volume is not mentioned in the caption.*
  Done.
- *Fig 9: no legend: values of the water height and of the mesh refinement ?*
  These images were created by the code during the simulation. We just specify the min and max values and the code builds a linear color scale from them. We added the min and max values in the legend.
- *Fig 10: Impossible to read the values when the article is printed, enlarge the figure ?*
  Done.
- *Typo : "ant" in the caption.*
  Done
- *page 12 line 6: "time delay" instead of "delay time".*

Replaced by "arrival time delay"

- *Fig 12 : the names of the gauges are the same as fig 7 but they are not located at the same places.*
  This is the choice of the experimenters. See their article for more information.
- *Fig 13: the volume is not mentioned in the caption.*
  Done.
- *page 13 line 15: sentence which does not contain a verb.*
  Corrected
- *page 14 line 1: refer to section 2.2 for the abrupt topography.*
  Done.
- *page 14 line 4: a "s" is missing in "figures" or change "these".*
  Done.
- *page 14 line 27: the euros (dollars ?) symbol must be changed twice.*
  Done.
- *Fig 14: no legend: values of the water height and of the mesh refinement ?*
  Done.
- *Fig 15: Impossible to read the values when the article is printed, enlarge the figure ?*
  The figure reads well on the pdf. This may be due to the configuration of the printer.
- *The time delay is not mentioned in the caption.*
  Done.
- *Fig 17: Impossible to read the legend when the article is printed, enlarge the figure ?*
  The figure reads well on the pdf. This may be due to the configuration of the printer.
- *page 17 line 18 : refer to section 2.3.4 for the threshold.*
  Done
- *page 22 line 5 : "B-flood" to change into "b-flood" ? The link is not correct, one must change the B to the lower case. Also add the first names of the authors.*
  Done

---

## Author Comment (AC2)

We would like to thank the referee whose comments helped to improve our article. We answer point by point to all these remarks, and have modified our article accordingly when necessary.

*On the CFL condition (Section 2.1, equation 8 in the manuscript) :*
- *First, I'm not sure that the velocity u = q/h has been introduced before.*

Done

- *Why do the authors not use the less restrictive CFL condition*
- *What motivates the choice instead of the classical choice*

It was a mistake, we have corrected it in accordance with the referee's remark.

- *What is the form of the CFL condition for the 2d computations ?*

It is the one for 2D thanks to the 0.5 prefactor

*On the computation of the source term (Section 2.2 in the manuscript) :*
- *The authors propose a second order accurate (in time and space) discretisation of the ux term but a firrst order in time discretization of the source term. Would it be possible to propose a stable second order in time discretisation of the source term, especially for the friction term, and then to obtain global second order accuracy in Figure 4 ?*

Indeed, this is a possibility. On the other hand, the characteristic time related to friction is much longer than the characteristic time of dynamics. The time step used is therefore much smaller than the one related to friction, and therefore this friction time does not affect the error, even if it is of order 1. For the cases testing convergence of the error, the error comes mainly from the edges of the domain.

- *Rain and inltration source terms are added in the mass equation only. How do the authors justify that they have no impact on the momentum equation ?*

The ratio between the momentum brought by the rain and the momentum of the flow is of the order of R/u. The characteristic velocities for very heavy rainfall R are of the order of 100mm/hour while the fluid velocities u are of the order of 1 m/s. We can reasonably neglect the momentum brought by the rain.

- *Equations (13) and (15) should be rewritten. The equality is valid in a code where we assign a value but not in a mathematical sense.*

Done

*On the comparison with analytical solutions and laboratory experiments (Section 3 in the manuscript) :*
- *Since it happens in flash flood events, it would be interesting to test the software agains an analyicial solution with dry areas, as the well known Thacker test case.*

This was done for the saint solver from basilisk, from which b-flood is derived. (http://www.basilisk.fr/src/test/parabola.c) You can find many other tests made on the Basilisk solver at this address : http://www.basilisk.fr/src/test/README#shallow-water-flows .

- *I do not understand what the sentence "For adaptive renement, the error threshold is set at 5 mm on the water level field" means. Does it mean that there is only two level of renement, one for value of h lower than the threshold and one for value of h greater than the threshold ? It does not seem to be the case on the numerical results, Figure 9.*

The 5 mm threshold is the error allowed between two level of refinement. You will find more explenation on the paragraph 2.4 : Adaptative refinement.

- *"Errors" defined by (17)-(18) are not errors since they can vanish for non identical solutions. The only errors here are the L2 errors defined by (19)-(20). Another word has to be used.*

Done

- *The words "and the presence of houses" at the end of Section 3.2.1 should be removed since the houses are introduced in the next section.*

The fluvial case also includes very small houses representing the real houses of the Toce Valley. You can see them disturbing the flow by zooming in on figure 9.

- *It would be interesting to give some snapshots of the mesh at different times to see the effect of the AMR strategy.*

We added the film as additional material.

- *"We record the maximal value of the water depth field during the entire event, as shown on gure 18 highlighting the flood extent." Is it possible to have access to a map of the flood as it happens in October 2015 or at least to some data at certain location that could be used for comparison with the numerical results ?*

This is the exact subject of another article that has already been published. We have added the reference at the end of the paragraph concerning the Cannes case. This article concerns other cities of the French Riviera because no data has been collected in Cannes (to our knowledge).

- *I know it is a huge work (probably out of the scope of this paper), but it would be interesting to analyse the sensibility of the results to the friction and inltration coefficients. In particular, a first attempt could be to reproduce the same event but with a larger high vegetation zone and to see if the results are signicantly dierent or not.*

Indeed, this type of study is beyond the scope of this article. The purpose of the article is to validate the b-flood code in the case of floods. We have added the case of Cannes to show that simulating a flash flood in a sufficiently short time is possible. Since b-flood is completely free, we hope that other teams will use our software to carry out studies like the one mentioned by the referee.